# Estimation of evapotranspiration by FAO Penman-Monteith Temperature and Hargreaves-Samani models under temporal and spatial criteria. A case study in Duero Basin (Spain).

Rubén Moratiel[1,2], Raquel Bravo[3], Antonio Saa[1,2], Ana M Tarquis[2], Javier Almorox[1]

[1]Department of Plant Production, Universidad Politécnica de Madrid, Avda. Complutense s/n, Madrid 28040, Spain

[2]CEIGRAM, Centro de Estudios e Investigación para la Gestión de Riesgos Agrarios y Medioambientales, C/Senda del Rey 13, Madrid, 28040, Spain

[3]Ministerio de Agricultura y Pesca, Alimentación y Medio Ambiente Paseo de la Infanta Isabel 1, Madrid, 28071, Spain

*Correspondence to:* Rubén Moratiel (ruben.moratiel@upm.es)

**Abstract.** Use of the Evapotranspiration based scheduling method is the most common one for irrigation programming in agriculture. There is no doubt that the estimation of the reference evapotranspiration ($ET_o$) is a key factor in irrigated agriculture. However, the high cost and maintenance of agrometeorological stations and high number of sensors required to estimate it creates a non-plausible situation especially in rural areas. For this reason, the estimation of $ET_o$ using air temperature, in places where wind speed, solar radiation and air humidity data are not readily available, is particularly attractive. Daily data record of 49 stations distributed over Duero basin (Spain), for the period 2000-2018, were used for estimation of $ET_o$ based on seven models against Penman-Monteith FAO 56 with temporal (annual or seasonal) and spatial perspective. Two Hargreaves-Samani models (HS), with and without calibration, and five Penman-Monteith temperature models (PMT) were used in this study. The results show that the models´ performance changes considerably depending on whether the scale is annual or seasonal. The performance of the seven models was acceptable at an annual perspective ($R^2 > 0.91$, NSE> 0.88, MAE <0.52 mm · $d^{-1}$ and RMSE <0.69 mm · $d^{-1}$). For winter, no model showed a good performance. In the rest of the seasons, the models with the best performance were three: $PMT_{CUH}$ [Penman-Monteith temperature with calibration of Hargreaves empirical coefficient ($k_{RS}$), average monthly value of wind speed and average monthly value of maximum and minimum relative humidity] , $HS_C$[Hargreaves-Samani with calibrarion of $k_{RS}$] and $PMT_{OUH}$ [Penman-Monteith temperature without calibration of $k_{RS}$, average monthly value of wind speed and average monthly value of maximum and minimum relative humidity]. $HS_C$ model presents a calibration of Hargreaves empirical coefficient ($k_{RS}$). In $PMT_{CUH}$ model, $k_{RS}$ was calibrated and average monthly values were used for wind speed, maximum and minimum relative humidity. Finally, $PMT_{OUH}$ model is as $PMT_{CUH}$ model except that $k_{RS}$ was not calibrated. These results

are very useful to adopt appropriate measures for an efficient water management, especially in the intensive agriculture in semi-arid zones, under the limitation of agrometeorological data.

40

## 1. Introduction

A growing population and its demand for food increasingly demand natural resources such as water. This, linked with the uncertainty of climate change, makes water management a key point for future food security. The main challenge is to produce enough food for a growing population that is directly affected by the challenges set in the management of agricultural water, mainly with irrigation management (Pereira, 2017).

Evapotranspiration (ET) is the water lost from the soil surface and surface leaves by evaporation and, by transpiration, from vegetation. ET is one of the major components of the hydrologic cycle and represented a loss of water from the drainage basin. Evapotranspiration (ET) information is key to understanding and managing water resources systems (Allen et al., 2011). ET is normally modeled using weather data and algorithms that describe aerodynamic characteristics of the vegetation and surface energy.

In agriculture, irrigation water is usually applied based on the water balance method in the soil water balance equation that allows the calculation of the decrease in soil water content as the difference between outputs and inputs of water to the field. In arid areas where rainfall is negligible during the irrigation season an average irrigation calendar may be defined a priori using mean ET values (Villalobos et al., 2016). The Food and Agricultural Organization of United Nations (FAO) improved and upgraded the methodologies for reference evapotranspiration ($ET_o$) estimation by introducing the reference crop (grass) concept, described by FAO Penman- Monteith (PM-$ET_o$) equation (Allen et al., 1998). This approach was tested well under different climates and time step calculations and is currently adopted worldwide (Allen et al., 1998, Todorovic et al., 2013; Almorox et al., 2015). To estimate crop evapotranspiration ($ET_c$) is obtained by function of two factor ($ET_c = K_c \cdot ET_o$): reference crop evapotranspiration ($ET_o$) and crop coefficient ($K_c$) (Allen et al. 1998). $ET_o$ was introduced to study the evaporative demand of the atmosphere independently of crop type, crop stage development and management practices. $ET_o$ is only affected by climatic parameters, and is computed from weather data. Crop influences are accounted for by using a specific crop coefficient ($K_c$). However, $K_c$ varies predominately with the specific crop characteristics and only to a limited extent with climate (Allen et al., 1998)

The ET is very variable locally and temporarily because of the climate differences. Because the ET component is relatively large in water hydrology balances any small error in its estimate or measurement represents large volumes of water (Allen et al., 2011). Small deviations in $ET_o$ estimations would affect irrigation and water management in rural areas in which crop extension is significant. For example, in 2017 there was a water shortage at the beginning of the cultivation period (March) at the Duero basin (Spain). The classical irrigated crops, i.e. corn, were replaced by others with lower water needs such as sunflower.

Wind speed (u), solar radiation (Rs), relative humidity (RH) and temperature (T) of the air are required to estimate $ET_o$. Additionally, vapor pressure deficit (VPD), soil heat flux (G) and net radiation (Rn) measurements or estimates are necessary. The PM-$ET_o$ methodology presents the disadvantage that required climate or weather data that are normally unavailable or low quality (Martinez and Thepadia, 2010) in rural areas. In this case, where data are missing, Allen et al. (1998) in the guidelines for PM-$ET_o$ recommend two approaches: a) using equation of Hargreaves-Samani (Hargreaves and Samani, 1985) and b) using PM temperature (PMT) method that requires data of temperature to estimate Rn (net radiation) and VPD for obtaining $ET_o$. In these situations, temperature-based evapotranspiration (TET) methods are very useful (Mendicino and Senatore, 2012).  Air temperature is the most available meteorological data, which are readily from most of climatic weather station. Therefore, TET methods and temperature databases are solid base for ET estimation all over the world including areas with limited data resources (Droogers and Allen, 2002).The first reference of the use of PMT for limited meteorological data was Allen (1995), subsequently, studies like those of Allen et al. (1996), Annandale et al. (2002), were carried out with similar behavior to HS and FAO-PM, although there was the disadvantage of a greater preparation and computation of the data than the HS method. On this point, it should be noticed that the researchers do not favor to using PMT formulation and adopting the HS equation, simpler and easier to use (Paredes et al., 2018). Authors like Pandey et al. (2014) performed calibrations based on solar radiations coefficients in Hargreaves-Samani equations. Today, PMT calculation process is easily implemented with the new computers (Pandey and Pandey, 2016; Quej et al., 2019).

Todorovic et al., (2013) reported that, in Mediterranean hyper-arid and arid climates PMT and HS show a similar behavior and performance while for moist sub-humid areas the best performance was obtained by PMT method. This behavior was reported for moist sub-humid areas in Serbia (Trajovic, 2005). Several studies confirm this performance in a range of climates (Martinez and Thepadia, 2010; Raziei and Pereira, 2013; Almorox, et al., 2015; Ren et al., 2016). Both models (HS and PMT) improved when local calibrations are performed (Gavilán et al., 2006; Paredes et al., 2018).  These reduce the problem when wind speed and solar radiation are the major driving variables.

Studies in Spain comparing HS and PMT methodologies were studied in moist sub–humid  climate zones (Northern Spain) showing a better fit in PMT than in HS. (Lopez Moreno et al., 2009). Tomas-Burguera (2017) reported for the Iberian Peninsula a better adjustment of PMT than HS, provided that the lost values were filled by interpolation and not by estimation in the model of PMT.

Normally the calibration of models for $ET_o$ estimation is done from a spatial approach, calibrating models in the locations studied. Very few studies have been carried out to test models from the seasonal point of view, being the annual calibration being the most studied. Meanwhile spatial and annual approaches are of great interest for climatology and meteorology, for agriculture, seasonal or even monthly calibrations are relevant for crop (Nouri and Homaee, 2018). To improve accuracy of $ET_o$ estimations, Paredes et al., 2018 used the values of the calibration constants values in the models were derived for October-March and April-September semesters.

The aim of this study was to evaluate the performance of temperature models for the estimation of reference evapotranspiration against the FAO56 Penman Monteith, with a temporal (annual or seasonal)

and spatial perspective in the Duero basin (Spain). The models evaluated were two Hargreaves-Samani (HS), with calibration and without calibration and five Penman-Monteith temperature model (PMT) analyzing the contribution of wind speed, humidity and solar radiation in a situation of limited agrometeorological data.

120

## 2. Materials and Method

### 2.1 Description of the Study Area

The study focuses on the Spanish part of the Duero hydrographic basin. The international hydrographic Duero basin is the most extensive of the Iberian Peninsula with 98073 km$^2$, it includes the territory of the Duero river basin as well as the transitional waters of the Oporto Estuary and the associated Atlantic coastal ones (CHD, 2019). It is a shared territory between Portugal with 19214 km$^2$ (19.6 % of the total area) and Spain with 78859 km$^2$ (80.4%). The Duero river basin is located in Spain between the parallels 43º 5' N and 40º 10' N and the meridians 7º 4' W and 1º 50' W (Fig. 1). This basin is almost exactly with the so-called *Submeseta Norte*, an area with an average altitude of 700 m, delimited by mountain ranges with a much drier central zone that contains large aquifers, being the most important area of agricultural production. The Duero Basin belongs in its 98.4% to the Autonomous Community of *Castilla y Léon*. The 70% of the average annual precipitation is used directly by the vegetation or evaporated from surface, this represents 35.000 hm$^3$. The remaining (30%) is the total natural runoff. Mediterranean is the predominant climate. The 90% of surface is affected by summer drought conditions. The average annual values are: 12 ºC of temperature and 612 mm of precipitation. However, precipitation ranges from minimum values of 400 mm (South-Central area of the basin) to a maximum of 1800 mm in the northeast of the basin (CHD, 2019). According to Lautensach (1967), 30 mm is the threshold definition of a dry month. Therefore, between 2 and 5 dry periods can be found in the basin (Ceballos et al., 2004). Moreover, the climate variability, especially precipitation, exhibited in the last decade has decreased the water availability for irrigation in this basin (Segovia- Cardozo et al., 2019).

The Duero basin has 4 million hectares of rainfed crops and some 500,000 hectares irrigated that consumes 75% of the basin's water resources consumption. Barley (*Hordeum vulgare* L.) is the most important rainfed crop in the basin occupying 36% of the National Crop Surface followed by wheat (*Triticum aestivum* L.) with 30% (MAPAMA, 2019). Sunflower (*Helianthus annuus* L.) representing 30% of the National crop surface. This crop is mainly unirrigated (90%). Maize (*Zea mays* L.), alfalfa (*Medicago sativa* L.) and sugar beet (*Beta vulgaris* L. var. sacharifera) are the main irrigated crops. These crops representing 29 %, 30% and 68% of each National crop area, respectively. Finally, Vine (*Vitis vinifera* L.) fills 72000 ha being less than 10% irrigated. For the irrigated crops of the basin there are water allocations that fluctuate depending on the availability of water during the agricultural year and the type of crop. These values fluctuate from 1200-1400 m$^3$ / ha for vine up to 6400-7000 m$^3$/ha for maize and alfalfa. The use rates of the irrigation systems used in the basin are: 25 %, 68% and 7% for surface, sprinkler and drip irrigation respectively (*Plan Hidrológico*, 2019).

155

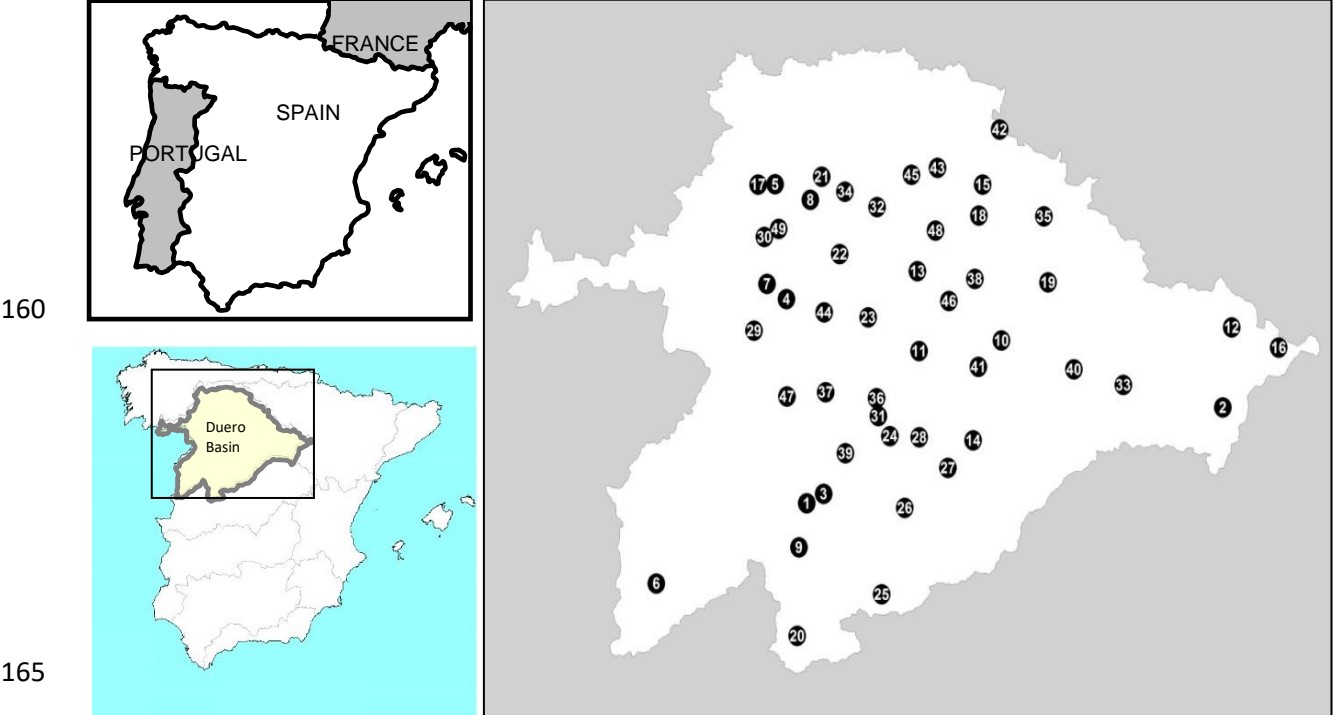

Figure. 1. Location of study area. The point with the number indicates the location of the agrometeorological stations according to Table 1.

170

## 2.2 Meteorological Data

The daily climate data were downloaded from 49 stations (Fig. 1B) from the agrometeorological network SIAR (Irrigation Agroclimatic Information System; SIAR in Spanish language), which is managed by the Spanish Ministry of Agriculture, Food and Environment (SIAR, 2018). The SIAR is coordinated by the Ministry of Agriculture, Fisheries and Food providing the basic meteorological data from weather stations distributed throughout the Duero Basin (Table 1). Each station incorporates measurements of air temperature (T) and relative humidity (RH; Vaisala HMP155), precipitation (ARG100 rain gauge), solar global radiation (pyranometer SKYE SP1110) and wind direction and wind speed (u) (wind vane and RM YOUNG 05103 anemometer). ). Sensors were periodically maintained and calibrated, and all data were recorded and averaged hourly on a data logger (Campbell CR10X and CR1000). Characteristics of the agrometeorological stations were described by (Moratiel et al., 2011, 2013a). For quality control, all parameters were checked, the sensors were periodically maintained and calibrated, all data being recorded and hourly averaged on a data logger.  The database calibration and maintenance are carried out by the Ministry of Agriculture. Transfer of data from stations to the Main Center is accomplished by modems; the Main Center incorporates a server, which sequentially connects to each station to download the information collected during the day. Once the data from the stations are downloaded, they are processed and transferred to a database. The Main Center is responsible for quality control procedures that comprise

the routine maintenance program of the network, including sensor calibration, checked for validity values and data validation. Moreover, the database was analyzed to find incorrect or missing values. To ensure that good quality data were used, we used quality control procedures to identify erroneous and suspect data. The quality control procedures applied are the range/limit test, step test consistency an internal test (Estevez et al., 2016).

The period studied was from 2000 to 2018, although the start date may fluctuate depending on the availability of data. Table 1 shows the coordinates of the agrometeorological stations used in the Duero Basin and the aridity index based on UNEP (1997). Table 1 the predominance of the semi-arid climate zone with 42 stations of the 49, being 2 arid, 4 dry-sub humid and 1 moist sub-humid.

Table 1. Agrometeorological station used in the study. Coordinates and Aridity Index.

| | Stations | Latitude [1] | Longitude [1] | Altitude (m) | Aridity Index |
|---|---|---|---|---|---|
| 1 | Aldearrubia | 40.99 | -5.48 | 815 | moist sub-humid |
| 2 | Almazán | 41.46 | -2.50 | 943 | semi-arid |
| 3 | Arabayona | 41.04 | -5.36 | 847 | semi-arid |
| 4 | Barcial del Barco | 41.93 | -5.67 | 738 | semi-arid |
| 5 | Bustillo del Páramo | 42.46 | -5.77 | 874 | semi-arid |
| 6 | Ciudad Rodrigo | 40.59 | -6.54 | 635 | semi-arid |
| 7 | Colinas de Trasmonte | 42.00 | -5.81 | 709 | semi-arid |
| 8 | Cubillas de los Oteros | 42.40 | -5.51 | 769 | semi-arid |
| 9 | Ejeme | 40.78 | -5.53 | 816 | semi-arid |
| 10 | Encinas de Esgueva | 41.77 | -4.10 | 816 | semi-arid |
| 11 | Finca Zamadueñas | 41.71 | -4.70 | 714 | semi-arid |
| 12 | Fuentecantos | 41.83 | -2.43 | 1063 | semi-arid |
| 13 | Fuentes de Nava | 42.08 | -4.72 | 744 | semi-arid |
| 14 | Gomezserracín | 41.30 | -4.30 | 870 | semi-arid |
| 15 | Herrera de Pisuerga | 42.49 | -4.25 | 821 | semi-arid |
| 16 | Hinojosa del Campo | 41.73 | -2.10 | 1043 | semi-arid |
| 17 | Hospital de Orbigo | 42.46 | -5.90 | 835 | semi-arid |
| 18 | Lantadilla | 42.34 | -4.28 | 798 | semi-arid |
| 19 | Lerma | 42.04 | -3.77 | 840 | semi-arid |
| 20 | Losar del Barco | 40.37 | -5.53 | 1024 | semi-arid |
| 21 | Mansilla mayor | 42.51 | -5.43 | 791 | semi-arid |
| 22 | Mayorga | 42.15 | -5.29 | 748 | semi-arid |
| 23 | Medina de Rioseco | 41.86 | -5.07 | 739 | semi-arid |
| 24 | Medina del Campo | 41.31 | -4.90 | 726 | arid |
| 25 | Muñogalindo | 40.58 | -4.93 | 1128 | arid |
| 26 | Nava de Arévalo | 40.98 | -4.78 | 921 | semi-arid |
| 27 | Nava de la Asunción | 41.17 | -4.48 | 822 | semi-arid |
| 28 | Olmedo | 41.31 | -4.69 | 750 | semi-arid |
| 29 | Pozuelo de Tábara | 41.78 | -5.90 | 714 | semi-arid |
| 30 | Quintana del Marco | 42.22 | -5.84 | 750 | semi-arid |

| | | | | | |
|---|---|---|---|---|---|
| 31 | Rueda | 41.40 | -4.98 | 709 | semi-arid |
| 32 | Sahagún | 42.37 | -5.02 | 856 | semi-arid |
| 33 | San Esteban de Gormaz | 41.56 | -3.22 | 855 | semi-arid |
| 34 | Santas Martas | 42.44 | -5.26 | 885 | semi-arid |
| 35 | Tardajos | 42.35 | -3.80 | 770 | dry sub-humid |
| 36 | Tordesillas | 41.49 | -5.00 | 658 | semi-arid |
| 37 | Toro | 41.51 | -5.37 | 650 | semi-arid |
| 38 | Torquemada | 42.05 | -4.30 | 868 | semi-arid |
| 39 | Torrecilla de la Orden | 41.23 | -5.21 | 793 | semi-arid |
| 40 | Vadocondes | 41.64 | -3.58 | 870 | semi-arid |
| 41 | Valbuena de Duero | 41.64 | -4.27 | 756 | semi-arid |
| 42 | Valle de Valdelucio | 42.75 | -4.13 | 975 | dry sub-humid |
| 43 | Villaeles de Valdavia | 42.56 | -4.59 | 885 | semi-arid |
| 44 | Villalpando | 41.88 | -5.39 | 701 | semi-arid |
| 45 | Villaluenga de la Vega | 42.53 | -4.77 | 927 | dry sub-humid |
| 46 | Villamuriel de Cerrato | 41.95 | -4.49 | 750 | dry sub-humid |
| 47 | Villaralbo | 41.48 | -5.64 | 659 | semi-arid |
| 48 | Villoldo | 42.27 | -4.59 | 817 | semi-arid |
| 49 | Zotes del Páramo | 42.26 | -5.74 | 779 | semi-arid |

[1] Degrees

## 2.3 Estimates of Reference Evapotranspiration

### 2.3.1 FAO Penman-Monteith (FAO-PM)

The FAO recommend the PM method as the one for computing $ET_o$ and evaluating other $ET_o$ models like the Penman-Monteith model using only temperature data (PMT) and other temperature-based model (Allen et al., 1998). The method estimates the potential evapotranspiration from a hypothetical crop with an assumed height of 0.12 m having aerodynamic resistance of ($r_a$) 208/$u_2$, ($u_2$ is the mean daily wind speed measured at a 2 m height over the grass) and a surface resistance ($r_s$) of 70 s·m$^{-1}$ and an albedo of

0.23, closely resembling the evaporation of an extension surface of green grass of uniform height, actively growing and adequately watered. The $ET_o$ (mm·d$^{-1}$) was estimated following FAO-56 (Allen et al. 1998):

$$ET_o = \frac{0.408\,\Delta\,(R_n - G) + \gamma\,\dfrac{900}{T+273}\,u_2(e_s - e_a)}{\Delta + \gamma(1 + 0.34u_2)} \qquad [1]$$

In Eq. 1, $R_n$ is net radiation at the surface (MJ m$^{-2}$ d$^{-1}$), G is ground heat flux density (MJ m$^{-2}$ d$^{-1}$), $\gamma$ is the psychrometric constant (kPa °C$^{-1}$), T is mean daily air temperature at 2 m height (°C), $u_2$ is wind speed at 2 m height (m s$^{-1}$), $e_s$ is the saturation vapor pressure (kPa), $e_a$ is the actual vapor pressure (kPa) and $\Delta$ is the slope of the saturation vapor pressure curve (kPa °C$^{-1}$). According to Allen et al. (1998) in Eq. 1, G can be considered as zero.

### 2.3.2 Hargreaves-Samani (HS)

The scarce availability of agrometeorological data (global solar radiation, air humidity and wind speed mainly) limit the use of the FAO-PM method in many locations. Allen et al., (1998) recommended applying Hargreaves–Samani expression for situations where only the air temperature is available. The Hargreaves-Samani formulation (HS) is an empirical method that requires empirical coefficients to calibrate(Hargreaves and Samani, 1982, 1985). The Hargreaves and Samani (Hargreaves and Samani, 1982, 1985) method is given by the following equation (2):

$$ET_o = 0.0135 \cdot k_{RS} \cdot 0.408 \cdot H_o \cdot (T_m + 17.8) \cdot (T_x - T_n)^{0.5} \qquad (2)$$

where $ET_o$ is the reference evapotranspiration (mm day$^{-1}$); $H_o$ is extraterrestrial radiation (MJ·m$^{-2}$·d$^{-1}$); $k_{RS}$ is the Hargreaves empirical coefficient, $T_m$, $T_x$ and $T_n$ are the daily mean, maximum and minimum air temperature (°C), respectively. The value $k_{RS}$ was initially set to 0.17 for arid and semiarid regions (Hargreaves and Samani, 1985). Hargreaves (1994) later recommended to use the value of 0.16 for interior regions and 0.19 for coastal regions. Daily temperature variations can occur due to other factors such as topography, vegetation, humidity, among others, thus using a fixed coefficient may lead to errors. In this study, we use the 0.17 as original coefficient (HS$_o$) and the calibrated coefficient $k_{RS}$ (HS$_c$).The $k_{RS}$ reduces the inaccuracy and consequently thus improving the estimation of ET$_o$. This calibration was done for each station.

### 2.3.3    Penman- Monteith Temperature (PMT)

The FAO-PM, when applied using only measured temperature data is denominated to as Penman-Monteith Temperature (PMT) retains many of the dynamics of the full data FAO-PM (Pereira et al., 2015; Hargreaves and Allen, 2003). Humidity and solar radiation are estimated in the PMT model using only air temperature as input for the calculation of ET$_o$. Wind speed in the PMT model is set as constant value of 2m/s  (Allen et al.1998).  In this model, where global solar radiation (or sunshine data) is lacking, the difference between the maximum and minimum temperature can be used, as an indicator of cloudiness and atmospheric transmittance, for the estimation of solar radiation [Eq.3] (Hargreaves and Samani, 1982). Net solar shortwave and longwave radiation estimates are obtained as indicated by Allen et al., (1998), equation 4 and 5 respectively. The expression of PMT is obtained as indicated in equations 4, 5, 6, 7 and 8.

$$R_s = H_o \cdot k_{RS} \cdot (T_x - T_n)^{0.5} \qquad (3)$$

$$R_{ns} = 0.77 \cdot H_o \cdot k_{RS} \cdot (T_x - T_n)^{0.5} \qquad (4)$$

where Rs is solar radiation (MJ·m$^{-2}$·d$^{-1}$) ; R$_{ns}$ is net solar shortwave radiation (MJ·m$^{-2}$·d$^{-1}$); $H_o$ is extraterrestrial radiation (MJ·m$^{-2}$·d$^{-1}$); $H_o$ was computed as a function of site latitude, and solar angle and the day of the year (Allen et al., 1998).  $T_x$ is daily maximum air temperature (°C), $T_n$ is daily minimum air temperature (°C). For $k_{RS}$ Hargreaves (1994) recommended to use $k_{RS}$ = 0.16 for interior regions and $k_{RS}$ = 0.19 for coastal regions. For better accuracy the coefficient $k_{RS}$ can be adjusted locally (Hargreaves

and Allen 2003). In this study two assumptions of $k_{RS}$ were made, one where a value of 0.17 was fixed and another where it was calibrated for each station.

$$R_{nl} = \left(1.35 \cdot \left(\frac{k_{RS} \cdot (T_x - T_n)^{0.5}}{0.75 - 2z10^{-5}}\right) - 0.35\right) \cdot \left(0.34 - 0.14\left(0.6108 \cdot exp\left(\frac{17.27 \cdot T_d}{T_d - 237.3}\right)\right)^{0.5}\right) \cdot \sigma \cdot$$
$$\left(\frac{(Tx+273.15)^4 + (Tn+273.15)^4}{2}\right) \qquad (5)$$

Where $R_{nl}$ is net longwave radiation (MJ·m$^{-2}$·d$^{-1}$) $T_x$ is daily maximum air temperature (ºC); $T_n$ is daily minimum air temperature (ºC); $T_d$ is dew point temperature (ºC) calculated with the $T_n$ according to Todorovic et al., 2013; $\sigma$ Stefan-Boltzmann constant for a day ($4.903 \cdot 10^{-9}$ MJ K$^{-4}$ m$^{-2}$ d$^{-1}$); z is the altitude (m).

$$PMT_{rad} = \left(\frac{0.408\Delta}{\Delta + \gamma(1 + 0.34u_2)}\right) \cdot (R_{ns} - R_{nl} - G) \qquad (6)$$

$$PMT_{aero} = \frac{\gamma \cdot \frac{900 \cdot u_2}{T_m + 273} \cdot \left(\left(\frac{e_s(T_x) + e_s(T_n)}{2}\right) - e_s(T_d)\right)}{\Delta + \gamma(1 + 0.34u_2)} \qquad (7)$$

$$PMT = PMT_{rad} + PMT_{aero} \qquad (8)$$

Where *PMT* is the reference evapotranspiration estimate by Penman-Monteith temperature method (mm·d$^{-1}$); *PMT$_{rad}$* is the radiative component of PMT (mm·d$^1$); *PMT$_{aero}$* is the aerodynamic component of *PMT* (mm·d$^{-1}$); $\Delta$ is the slope of the saturation vapor pressure curve (kPa °C$^{-1}$), $\gamma$ is the psychrometric constant (kPa °C$^{-1}$), $R_{ns}$ is net solar shortwave radiation (MJ m$^{-2}$d$^{-1}$), $R_{nl}$ is net longwave radiation (MJ m$^{-2}$d$^{-1}$), *G* is ground heat flux density (MJ m$^{-2}$ d$^{-1}$) considered zero according to Allen et al.1998 , $T_m$ is mean daily air temperature (°C), $T_x$ is maximum daily air temperature, $T_n$ is mean daily air temperature, $T_d$ is dew point temperature (ºC) calculated with the $T_n$ according to Todorovic et al. (2013), $u_2$ is wind speed at 2 m height (m s$^{-1}$) and $e_s$ is the saturation vapor pressure (kPa). In this model two assumptions of $k_{RS}$ were done, one where a value of 0.17 was fixed and another where it was calibrated for each station.

### 2.3.4    Calibration and models

We studied two methods to estimate the ETo: Hargreaves–Samani (HS) and reference evapotranspiration estimate by Penman-Monteith temperature (PMT). Within these methods, different adjustments are proposed based on the adjustment coefficients of the methods and the missing data. The parametric calibration for the 49 stations was applied in this study. In order to decrease the errors of the evapotranspiration estimates, local calibration was used. The seven methods used with the coefficient

($k_{RS}$) of the calibrated and characteristics in the different locations studied are showed in Table 2. The calibration of the model coefficients was achieved by the nonlinear least squares fitting technique. The analyzed were calculated on yearly and seasonal bases. The seasons were the following: (1) winter (December, January, and February or DJF), (2) spring (March, April, and May or MAM), (3) summer (June, July, and August or JJA), (4) autumn (September, October, and November or SON).

Table 2. Characteristics of the models used in this study.

| Model | Coefficient $k_{RS}$ | $u_2$ (m/s) | Td (ºC) |
|---|---|---|---|
| $HS_O$ | 0.17 | - | - |
| $HS_C$ | Calibrated | - | - |
| $PMT_{O2T}$ | 0.17 | 2 | Todorovic[1] |
| $PMT_{C2T}$ | Calibrated | 2 | Todorovic[1] |
| $PMT_{OUT}$ | 0.17 | Average[2] | Todorovic[1] |
| $PMT_{OUH}$ | 0.17 | Average[2] | Average[3] |
| $PMT_{CUH}$ | Calibrated | Average[2] | Average[3] |

[1]Dew point temperature obtained according to Todorovic et al. (2013).
[2]Average monthly value of wind speed
[3]Average monthly value of maximum and minimum relative humidity.

### 2.4. Performance assessment.

Model´s suitability, accuracy and performance were evaluated using coefficient of determination ($R^2$; Eq. [9]) of the n pairs of observed ($O_i$) and predicted ($P_i$) values. Also, the mean absolute error (MAE, mm·d$^{-1}$; Eq. [10]), root mean square error (RMSE; Eq. [11]) and The Nash-Sutcliffe model efficiency coefficient (NSE; Eq. [12]) (Nash and Sutcliffe 1970) was used. The coefficient of regression line (b), forced through the origin, is obtained by predicted values divided by observed values ($ET_{model}/ET_{FAO56}$)
The results were represented in a map applying of the Kriging method with the Surfer® 8 program.

$$R^2 = \left\{ \frac{\sum_{i=1}^{n}(O_i - \bar{O}) \cdot (P_i - \bar{P})}{[\sum_{i=1}^{n}(O_i - \bar{O})^2]^{0.5} \cdot [\sum_{i=1}^{n}(P_i - \bar{P})^2]^{0.5}} \right\}^2 \quad (9)$$

$$MAE = \frac{1}{n}\sum_{i=1}^{n}(|O_i - P_i|) \; (mm.d^{-1}) \quad (10)$$

$$RMSE = \left[ \frac{\sum_{i=1}^{n}(O_i - P_i)^2}{n} \right]^{0.5} (mm.d^{-1}) \quad (11)$$

$$NSE = 1 - \left[ \frac{\sum_{i=1}^{n}(O_i - P_i)^2}{\sum_{i=1}^{n}(O_i - \bar{O})^2} \right] \quad (12)$$

 **3. Results and Discussion**

In the study period the data indicated that the Duero basin is characterized by being a semiarid climate zone (94% of the stations) where the $P / ET_o$ ratio is between 0.2-0.5 (Todorovic et al., 2013). The mean annual rainfall is 428 mm while the average annual $ET_o$ for Duero basin is of 1079 mm, reaching the maximum values in the zone center-south with values that surpass slightly 1200 mm (Fig. 2). The great

temporal heterogeneity is observed in the Duero Basin with values of 7% of the $ET_o$ during the winter months (DJF) while during the summer months (JJA) they represent 47% of the annual $ET_o$. In addition, the months from May till September represent 68% of the annual $ET_o$, with similar values as reported by Moratiel et al. (2011).

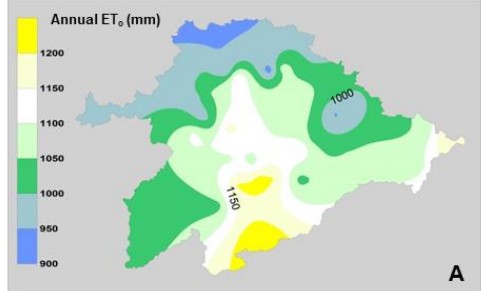

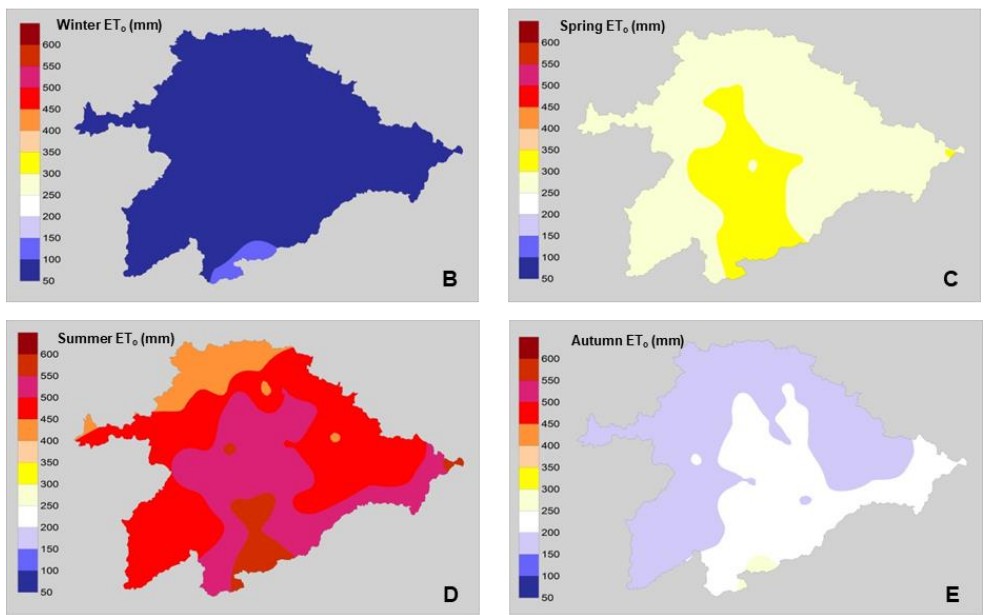

Figure.2. Mean values season of $ET_o$ (mm) during the study period 2000-2018. A, annual; B, winter (December, January, and February or DJF); C, spring (March, April, and May or MAM); D, summer (June, July, and August or JJA) and E, autumn (September, October, and November or SON).

Table 3 shows the different statistics analyzed in the seven models studied as a function of the season of the year and annually. From an annual point of view all the models show $R^2$ values higher than 0.91, NSE higher than 0.88, MAE less than 0.52, RMSE lower than 0.69 and underestimates or overestimates of the models by ±4%. The best behaviour is shown by the $PMT_{CHU}$ model with MAE and RMSE of 0.39 mm·d$^{-1}$ and 0.52 mm·d$^{-1}$ respectively. $PMT_{CUH}$ shows no tendency to overestimate or underestimate the values

in which it is observed a coefficient of regression b of 1.0. This model show values of NSE and $R^2$ of 0.93. The models HSc and $PMT_{OUH}$ have a similar behavior with same MAE (0.41 mm·d$^{-1}$), NSE (0.92) and $R^2$ (0.91). RMSE is 0.55 mm·d$^{-1}$ for $PMT_{OUH}$ model and 0.54 mm·d$^{-1}$ for HSc model. The models $PMT_{OUT}$ and HSo showed a slightly higher performance than $PMT_{O2T}$ and $PMT_{C2T}$, being these last two models the worst behaviors showed (Fig.3). The performance of the models ($PMT_{O2T,}$ $PMT_{OUT}$ and

$PMT_{OUH}$] improve as the averages of wind speed (u) and dew temperature ($T_d$) values are incorporated. The same pattern is shown between the $PMT_{CUH}$ models, where the mean u values and $T_d$ are incorporated, and $PMT_{C2T}$, with u of 2 m/s and dew temperature with the approximation of Todorovic et al. (2013). These adjustments are supported because the adiabatic component of evapotranspiration in the PMT equation is very influential in the Mediterranean climate, especially wind speed (Moratiel et al.,

2010).

From a spatial perspective, it is observed in Fig. 3 that the areas where the values of MAE are higher are to the east and southwest of the basin. This is due to the fact that the average wind speed in the eastern zone is higher than 2.5 m/s, for example, the Hinojosa del Campo station shows average annual values of 3.5 m/s. The southwest area shows values of wind speeds below 1.5 m/s as the Ciudad Rodrigo station

with annual average values of 1.19 m/s.

Table 3. Statistical indicators for ET$_o$ estimation in the seven models studied for different season. Average data for the 49 stations studied.

| Season | Variable | MODEL | | | | | | | Daily Average (ETo FAO56, mm·d$^{-1}$) |
|---|---|---|---|---|---|---|---|---|---|
| | | HS$_O$ | HS$_C$ | PMT$_{O2T}$ | PMT$_{C2T}$ | PMT$_{OUT}$ | PMT$_{OUH}$ | PMT$_{CUH}$ | |
| Annual | $R^2$ | 0.93 | 0.93 | 0.91 | 0.91 | 0.92 | 0.93 | 0.93 | |
| | NSE | 0.90 | 0.92 | 0.88 | 0.89 | 0.90 | 0.92 | 0.93 | |
| | MAE (mm·d$^{-1}$) | 0.47 | 0.41 | 0.52 | 0.50 | 0.47 | 0.41 | 0.39 | 2.95 |
| | RMSE(mm·d$^{-1}$) | 0.62 | 0.54 | 0.69 | 0.66 | 0.62 | 0.55 | 0.52 | |
| | RMSE (%) | 21.0 | 18.5 | 23.4 | 22.3 | 20.9 | 18.7 | 17.8 | |
| | b | 1.03 | 0.97 | 1.04 | 1.02 | 1.03 | 1.03 | 1.00 | |
| Winter (DJF) | $R^2$ | 0.53 | 0.53 | 0.56 | 0.55 | 0.56 | 0.59 | 0.59 | |
| | NSE | 0.43 | 0.50 | 0.36 | 0.35 | 0.35 | 0.57 | 0.58 | |
| | MAE (mm·d$^{-1}$) | 0.27 | 0.25 | 0.29 | 0.30 | 0.30 | 0.24 | 0.24 | 0.90 |
| | RMSE(mm·d$^{-1}$) | 0.35 | 0.33 | 0.36 | 0.37 | 0.37 | 0.30 | 0.30 | |
| | RMSE (%) | 38.3 | 36.1 | 40.3 | 40.5 | 41.2 | 33.6 | 33.5 | |
| | b | 0.99 | 0.93 | 1.07 | 1.06 | 1.09 | 0.96 | 0.96 | |
| Spring (MAM) | $R^2$ | 0.83 | 0.83 | 0.81 | 0.81 | 0.81 | 0.82 | 0.82 | |
| | NSE | 0.80 | 0.81 | 0.75 | 0.78 | 0.74 | 0.80 | 0.81 | 3.19 |
| | MAE (mm·d$^{-1}$) | 0.43 | 0.42 | 0.50 | 0.46 | 0.52 | 0.45 | 0.43 | |
| | RMSE(mm·d$^{-1}$) | 0.56 | 0.55 | 0.62 | 0.59 | 0.65 | 0.57 | 0.55 | |

| | | | | | | | | |
|---|---|---|---|---|---|---|---|---|
| | RMSE(%) | 17.5 | 17.2 | 19.6 | 18.4 | 20.2 | 18.0 | 17.3 | |
| | b | 1.01 | 0.95 | 1.04 | 1.00 | 1.06 | 1.02 | 0.99 | |
| Summer (JJA) | $R^2$ | 0.59 | 0.59 | 0.53 | 0.53 | 0.56 | 0.60 | 0.60 | |
| | NSE | 0.32 | 0.54 | 0.21 | 0.31 | 0.45 | 0.52 | 0.59 | |
| | MAE (mm·d$^{-1}$) | 0.68 | 0.56 | 0.72 | 0.68 | 0.62 | 0.57 | 0.53 | 5.48 |
| | RMSE(mm·d$^{-1}$) | 0.84 | 0.71 | 0.91 | 0.87 | 0.79 | 0.73 | 0.68 | |
| | RMSE(%) | 15.4 | 13.0 | 16.6 | 15.8 | 14.4 | 13.3 | 12.3 | |
| | b | 1.04 | 0.98 | 1.03 | 1.00 | 1.00 | 1.03 | 1.00 | |
| Autumn (SON) | $R^2$ | 0.85 | 0.85 | 0.83 | 0.83 | 0.84 | 0.86 | 0.86 | |
| | NSE | 0.72 | 0.82 | 0.61 | 0.65 | 0.78 | 0.83 | 0.85 | |
| | MAE (mm·d$^{-1}$) | 0.50 | 0.40 | 0.58 | 0.55 | 0.46 | 0.40 | 0.38 | 2.21 |
| | RMSE(mm·d$^{-1}$) | 0.62 | 0.52 | 0.73 | 0.70 | 0.58 | 0.51 | 0.49 | |
| | RMSE(%) | 28.1 | 23.5 | 32.8 | 31.6 | 26.2 | 23.1 | 22.1 | |
| | b | 1.09 | 1.02 | 1.14 | 1.12 | 1.07 | 1.05 | 1.02 | |

These MAE differences are more pronounced in those models in which the average wind speed is not taken, such as the $PMT_{C2T}$ and $PMT_{O2T}$ models. Most of the basin takes values of wind speeds between 1.5 and 2.5 m/s. The lower MAE values in the northern zone of the basin are due to the lower average values of vapour pressure deficit (VPD) than the central area, with values of 0.7 kPa in the northern zone

and 0.95 kPa in the central zone. Same trends in the effect of wind on the ETo estimates were detected by Nouri and Homaee (2018) where they indicated that values outside the range of 1.5-2.5 m/s in models where the default u was set at 2 m / s, increased the error of the ETo. Even models such as HS, where the influence of the wind speed values are not directly indicated outside the ranges previously mentioned, their performance is not good and some authors have proposed HS calibrations based on wind speeds in

Spanish basins such as the Ebro Basin (Martinez-Cob and Tejero-Juste, 2004). In our study, the HSc model showed good performance with MAE values similar to $PMT_{CUH}$ and $PMT_{OUH}$ (Fig.3). The performance of the models by season of the year changes considerably, obtaining lower adjustments with values of $R^2$ 0.53 for winter (DJF) in the models of HSo and HSc and for summer (JJA) in the models $PMT_{O2T}$ and $PMT_{C2T}$. All models during spring and autumn show $R^2$ above 0.8. The NSE for models $HS_O$,

$PMT_{C2T}$, $PMT_{O2T}$ and $PMT_{OUT}$ in summer and winter are at unsatisfactory values below 0.5 (Moriasi et al. 2007). The mean values (49 stations) of MAE and RMSE for the models in the winter were 0.24 -0.30 mm·d$^{-1}$ and 0.3-0.37 mm·d$^{-1}$ respectively. For spring, the ranges were between 0.42-0.52 mm·d$^{-1}$ for MAE and 0.55-0.65 mm·d$^{-1}$ for RMSE. In summer, MAE fluctuated between 0.53-0.72 mm·d$^{-1}$ and RMSE 0.68-0.91 mm·d$^{-1}$. Finally, in autumn, the values of MAE and RMSE were 0.38-0.58 and 0.49-

0.70 mm·d$^{-1}$ respectively (Table 3).

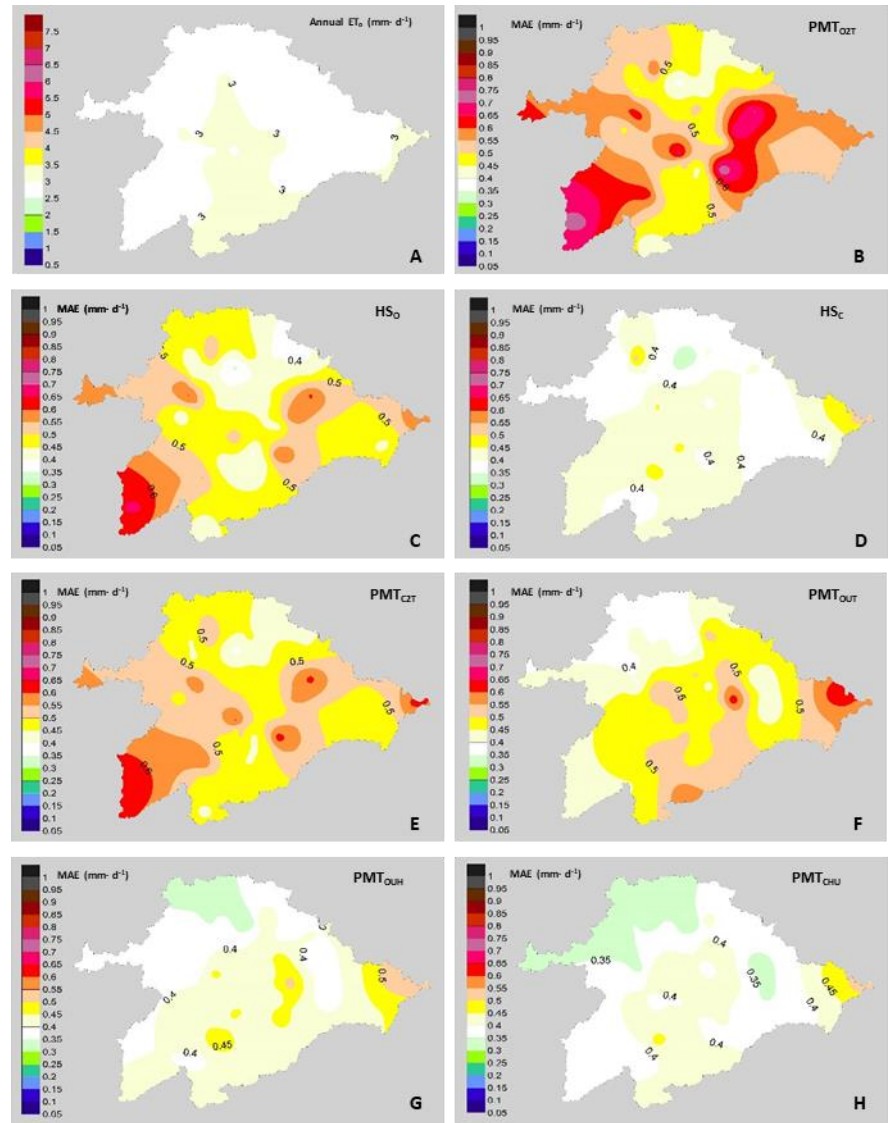

Figure 3. Performance of the models with an annual focus. A, Average annual values of $ET_o$ (mm·d$^{-1}$). Mean values of MAE (mm·d$^{-1}$): B, PMT$_{O2T}$ model; C, H$_O$ model; D, H$_C$ model; E, PMT$_{C2T}$ model; F, PMT$_{OUT}$ model; G, PMT$_{OUH}$ model and H, PMT$_{CUH}$ model

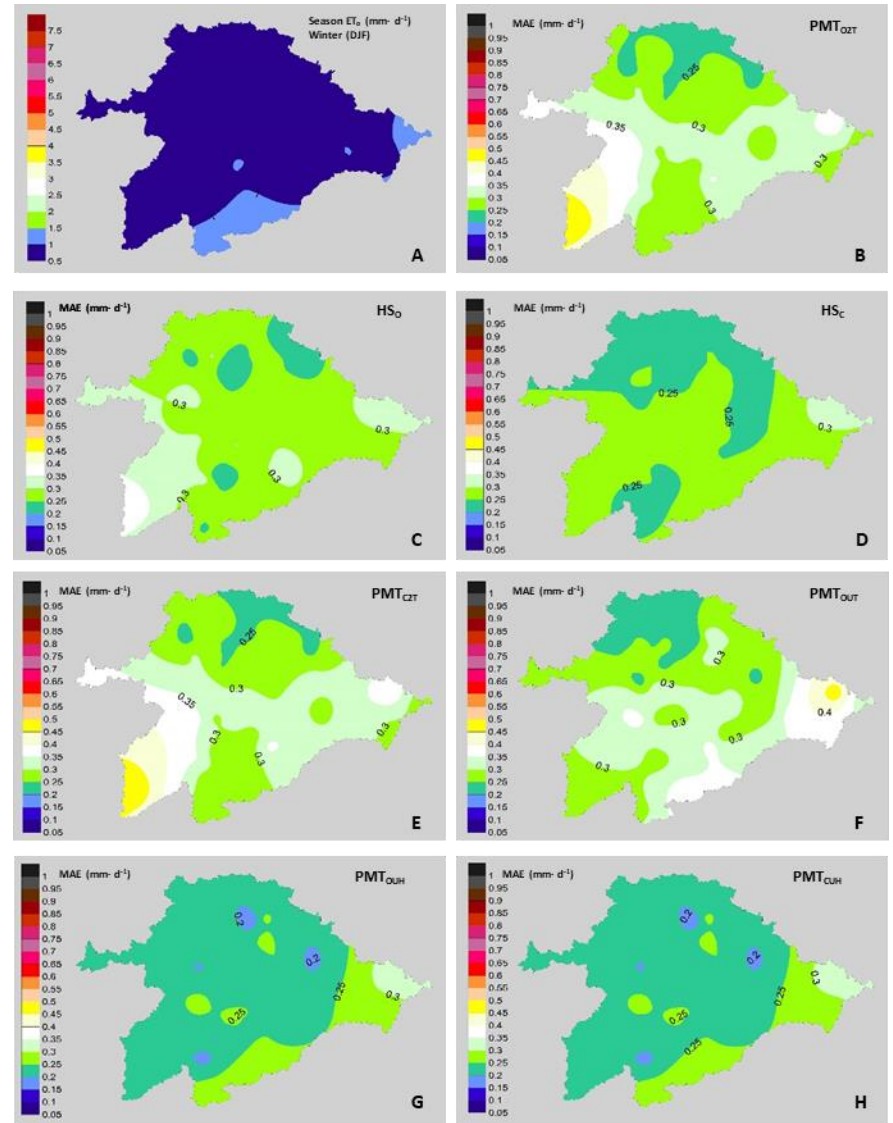

Figure 4. Performance of the models with a winter focus (December, January and February). A, Average values of ET$_o$ (mm·d$^{-1}$) in winter. Mean values of MAE (mm·d$^{-1}$):  B, PMT$_{O2T}$ model; C, H$_O$ model; D, H$_C$ model; E, PMT$_{C2T}$ model; F, PMT$_{OUT}$ model; G, PMT$_{OUH}$ model  and  H, PMT$_{CUH}$ model

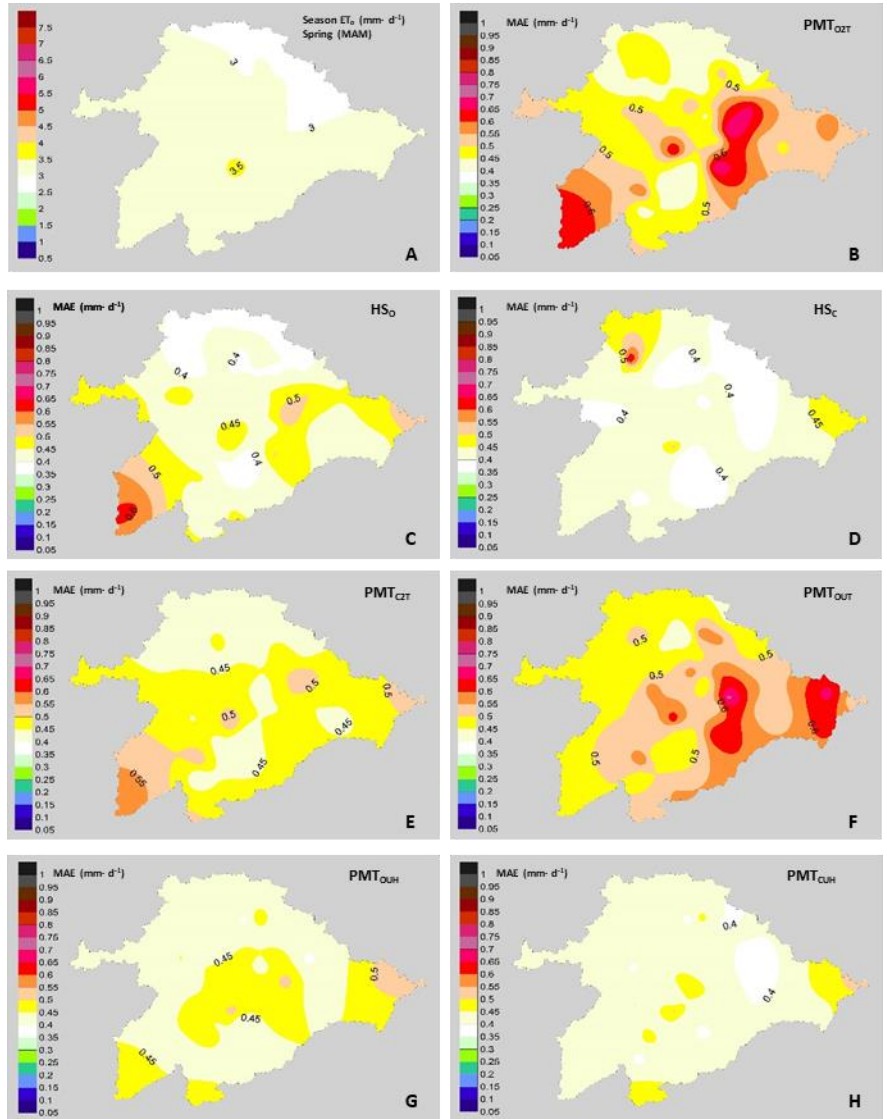

Figure.5. Performance of the models with a spring focus (March, April and May). A, Average annual values of $ET_o$ (mm·d$^{-1}$) in spring. Mean values of MAE (mm·d$^{-1}$):  B, PMT$_{O2T}$ model; C, H$_O$ model; D, H$_C$ model; E, PMT$_{C2T}$ model; F, PMT$_{OUT}$ model; G, PMT$_{OUH}$ model  and  H, PMT$_{CUH}$ model

The model that shows the best performance independently of the seasonal is the PMT$_{CUH}$. The models that can be considered in a second level are the HS$_C$ and the PMT$_{OUH}$. During the months of more solar radiation (summer and spring) the performance of the HS$_C$ model is slightly better than the PMT$_{OUH}$ model. The following models: HSo, PMTO2T, PMTC2T and PMTOUT, have a much lower performance than the previous models (PMT$_{OUH}$ and HS$_C$). The model that has the worst performance is the PMT$_{O2T}$..

The northern area of the basin is the area in which lower MAE shows in most models and for all seasons. This is due in part to the fact that the lower values of $ET_o$ (mm·d$^{-1}$) are located in the northern zone. On the other hand, the eastern zone of the basin shows the highest values of MAE error due to the strong winds that are located in that area.

During the winter the seven models tested show no great differences between them, although the $PMT_{CUH}$ is the model with the best performance. It is important to indicate that during this season the RMSE (%) is placed in all the models above 30%, so they can be considered as very weak models. According to Jamieson et al. (1991) and Bannayan and Hoogenboom (2009) the model is considered excellent with a normalized RMSE (%) less than 10%, good if the normalized RMSE (%) is greater than 10 and less than 20%, fair if the normalized RMSE (%) is greater than 20% and less than 30%, and poor if the normalized RMSE (%) is greater than 30%. All models that are made during the spring season (MAM) can be considered as good / fair since their RMSE (%) fluctuates between 17-20%. The seven models that are made during summer season (JJA) can be considered as good since their RMSE varies from 12 to 16%. Finally, the models that are made during autumn (SON) are considered fair / poor fluctuating between the values of 22-32%. The models that reached values greater than 30% during autumn were the model $PMT_{C2T}$ (31%) and $PMT_{O2T}$ (32%) also with a clear tendency to overestimation (Table 3) In the use of temperature models for estimating $ET_o$, it is necessary to know the objective that is set. For the management of irrigation in crops is better to test the models in the period in which the species require the contribution of additional water. In many cases applying the models with an annual perspective with a good performance can lead to more accentuated errors in the period of greater water needs. The studies of different temporal and spatial scales of the temperature models for $ET_o$ estimation, can give valuable information that allow to manage the water planning in zones where the economic development does not allow the implementation of agrometeorological stations due to its high cost.

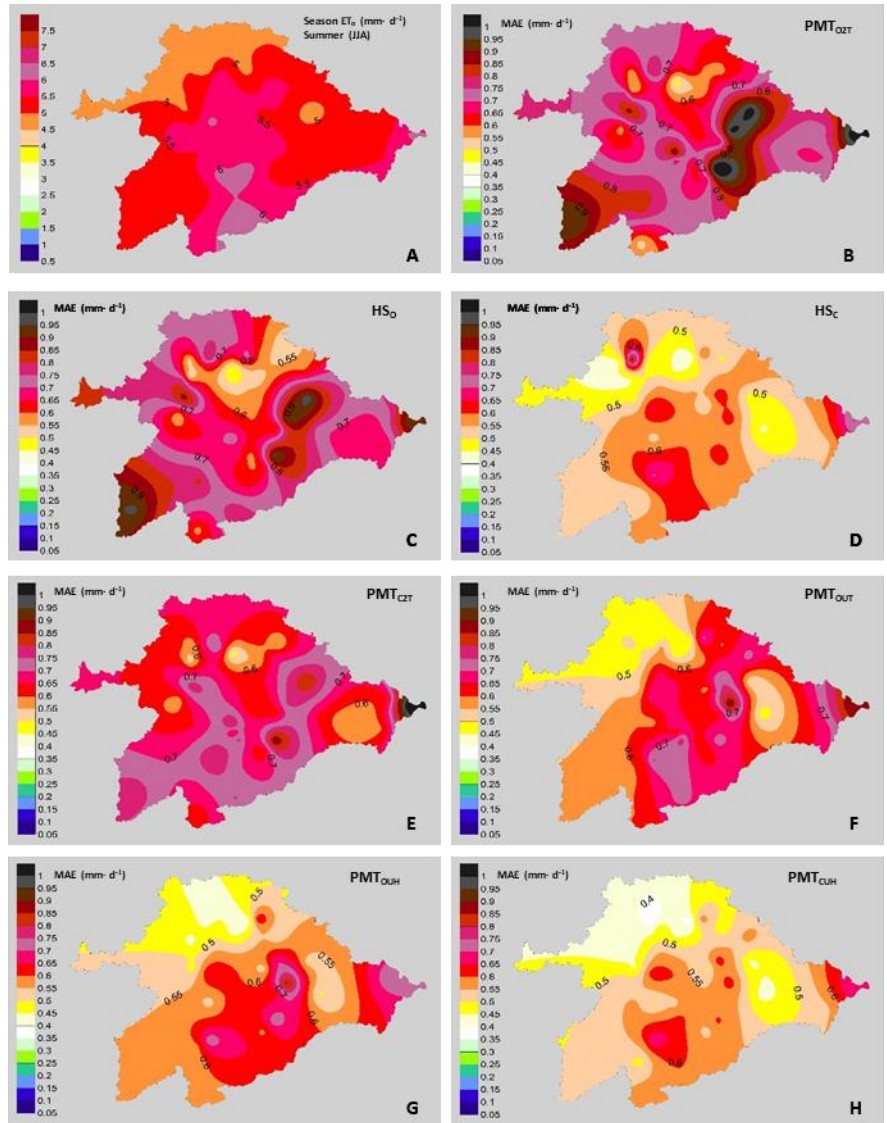

Figure.6. Performance of the models with a summer focus (June, July and August). A, Average values of $ET_o$ (mm·d$^{-1}$) in summer. Mean values of MAE (mm·d$^{-1}$): B, PMT$_{O2T}$ model; C, H$_O$ model; D, H$_C$ model; E, PMT$_{C2T}$ model; F, PMT$_{OUT}$ model; G, PMT$_{OUH}$ model and H, PMT$_{CUH}$ model

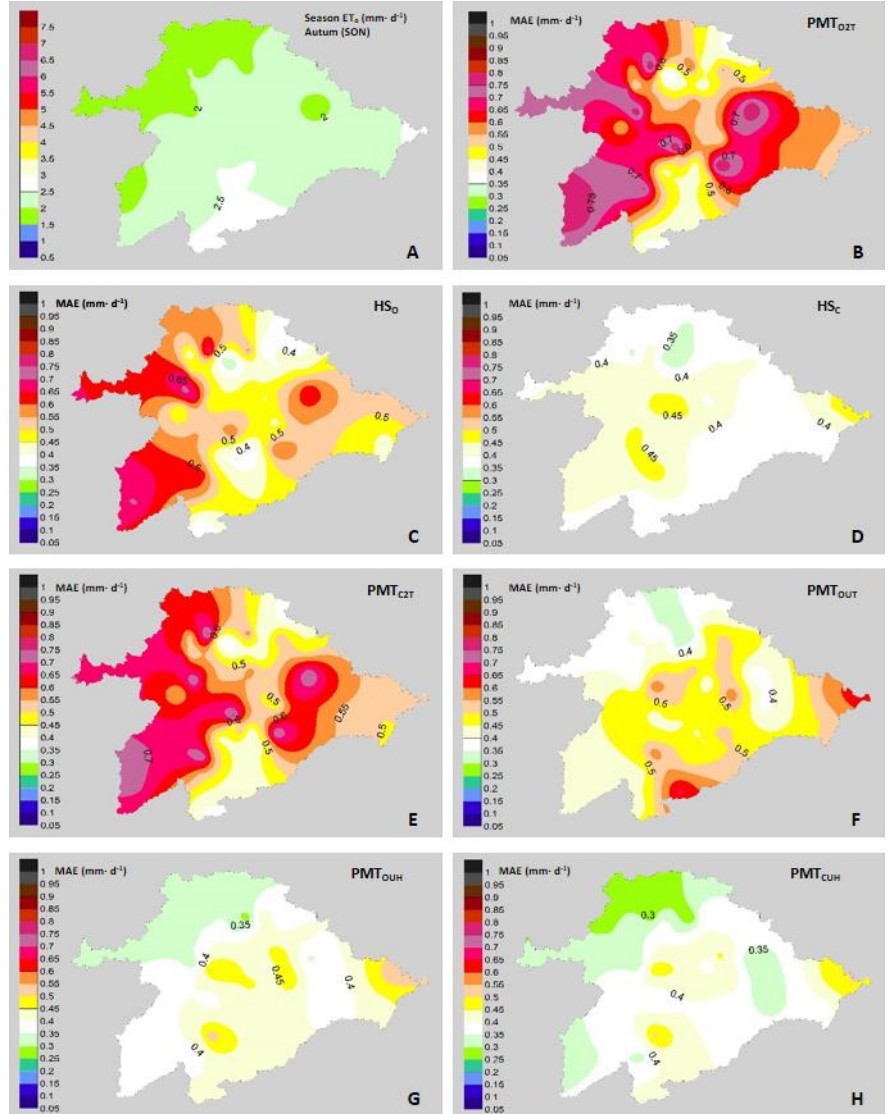

Figure.7. Performance of the models with an autumn focus (September, October and November). A, Average values of $ET_o$ (mm·d$^{-1}$) in autumn. Mean values of MAE (mm·d$^{-1}$):  B, PMT$_{O2T}$ model; C, H$_O$ model; D, H$_C$ model; E, PMT$_{C2T}$ model; F, PMT$_{OUT}$ model; G, PMT$_{OUH}$ model  and  H, PMT$_{CUH}$ model.

## 4.  Discussion

In annual seasons our data of RMSE fluctuates from 0.69 mm·d$^{-1}$ (PMT$_{O2T}$) to 0.52 mm·d$^{-1}$(PMT$_{O2T}$). These data are in accordance with the values cited by other authors in the same climatic zone.  Jabloun and Sahli (2008) cited RMSE of 0.41-0.80 mm·d$^{-1}$ for Tunisia. The authors showed the PMT model performance better than for the Hargreaves non calibrated model. Razand Pereira (2013) reported data of RMSE for semiarid zone in Iran between 0.27 and 0.81 mm·d$^{-1}$ for HSc model and 0.30 and 0.79 mmm·d$^{-1}$ for PMT$_{C2T}$, although these authors use monthly averages in their models. Ren et al. (2016) reported values of RMSE in a range of 0.51 to 0.90 mm·d-1 for PMT$_{C2T}$ and range of  0.81 to 0.94 mm·d$^{-1}$ for HSc in semiarid locations in Inner Mongolia (China). Todorovic et al. (2013) found that the PMT$_{O2T}$ method have better performance than the uncalibrated HS method (HS$_O$), with RMSE average of 0.47 mm·d$^{-1}$ for PMT$_{O2T}$ and 0.52 HS$_O$.  At this point, we should highlight that in our study daily values data have been used.

The original Hargreaves equation was developed by regressing cool season grass ET in Davis California, the $k_{RS}$ coefficient is a calibration coefficient. The Arity Index for Davis is semiarid (P/ET=0.33) (Hargreaves and Allen, 2003; Moratiel et al. 2013b) like 94% of the stations studied which explains why the behavior of the HSo model is often very similar to HSc. Even so, the calibration coefficient needs to be adjusted for other climates. Numerous studies in the literature have demonstrated the relevance of the $k_{RS}$ calibration model for estimating FAO56 (Todorovic et al., 2013, Raziei and Pereira, 2013, Paredes et al., 2018)

PMT models have improved considering the average wind speed. In addition, trends and fluctuations of $u$ have been reported as the factor that most influences $ET_o$ trends (Nouri et al., 2017, McVicar et al., 2012; Moratiel et al., 2011). Numerous authors have recommended to include, as much as possible, average data of local wind speeds for the improvement of the models as Nouri and Homaee (2018) and Raziei and Pereira (2013) in Iran, Paredes et al. (2018) in Azores islands (Portugal), Djaman et al. (2017) in Uganda, Rojas and Sheffield (2013) in Louisiana (USA), Jabloun and Shali (2008) in Tunisia and Martinez-Cob and Tejero-Juste (2004) in Spain, among others. In addition, even ETo prediction models based in PMT focus their behavior based on the wind speed variable (Yang et al., 2019). It is important to note that the $PMT_{OUT}$ generally has a better performance than the $PMT_{C2T}$ except for spring. The difference between both models is that in the $PMT_{C2T}$ $k_{RS}$ is calibrated with wind speed set at 2 m/s and in the $PMT_{OUT}$ $k_{RS}$ is not calibrated and with an average wind. In this case the wind speed variable affects less than the calibration of $k_{RS}$ since the average values of wind during spring (2.3 m/s) is very close to 2 m/s and there is no great variation between both settings. In this way, $k_{RS}$ calibration shows a greater contribution than the average of the wind speed to improve the model (Fig.5 E, F). In addition, although $u$ is not directly considered for HS, this model is more robust in regions with speed averages around 2 m/s (Allen et al. 1998 and Nouri and Homaee, 2018) On the other hand errors in the estimation of relative humidity cause substantial changes in the estimation of $ET_o$ as reported by Nouri and Homaee (2018) and Landeras et al. (2008).

The results of RMSE values (%) of the different models change considerably by seasons, values between 16.6% and 12.3% for summer and between 41.2% and 33.5% for winter. Similar results were obtained in Iran by Nouri and Homaee (2018), where the months of December-January and February the performance of the PMT and HS models tested had RMSE (%) values above 30%. Very few studies, as far as we know, have been carried out of adjustments of evapotranspiration models from a temporal point of view and generally the models are usually calibrated and adjusted from an annual point of view. Some authors, such as Aguilar and Polo (2011), differentiate seasons as wet and dry, others such as Paredes et al. (2018) divide in summer and winter, Vangelis et al. (2013) take into account two periods and Nouri and Homaee (2018) do it from a monthly point of view. In most cases, the results obtained in these studies are not comparable with those performed in this, since the time scales are different. However, it can be indicated that the results of the models according to the time scale season differ greatly with respect to the annual scale.

## 5. Conclusions

The performance of seven temperature-based models (PMT and HS) were evaluated in the Duero basin (Spain) with a total of 49 agrometeorological stations. Our studies revealed that the models tested on an annual or seasonal basis provide different performance. The values of $R^2$ are higher when they are performed annually with values between 0.91-0.93 for the seven models, but when performed from a seasonal perspective there are values that fluctuate between 0.5-0.6 for summer or winter and 0.86-0.81 for spring and autumn. The NSE values are high for models tested from an annual view, but for the seasons of spring and summer they are in values below 0.5 for the models $HS_O$, $PMT_{O2T}$, $PMT_{C2T}$ and $PMT_{OUT}$. The fluctuations between models with annual perspective of RMSE and MAE were greater than if those models were compared with a seasonal perspective. During the winter none of the models showed a good performance with values of $R^2 > 0.59$ NSE $> 0.58$ and RMSE (%) $> 30\%$. From a practical point of view in the management of irrigated crops, winter is a season where crop water needs are minimal with daily average values of ETo around 1 mm due to low temperatures, radiation and VPD. The model that showed the best performance was $PMT_{CUH}$ followed by $PMT_{OUH}$ and $HS_C$ for annual and seasonal criteria. $PMT_{OUH}$ is slightly less robust than $PMT_{CUH}$ during the maximum radiations periods of spring and summer since the $PMT_{CHU}$ performs the $k_{RS}$ calibration. The performance of the $HS_C$ model is better in the spring period, which is similar to $PMT_{CHU}$. The spatial distribution of MAE errors in the basin shows that it is highly dependent on wind speeds, obtaining greater errors in areas with winds greater than 2.8 m/s (east of the basin) and lower than 1.3 m/s (south-southwest of the basin). This information of the tested models in different temporal and spatial scales can be very useful to adopt appropriate measures for an efficient water management under limitation of agrometeorological data and under the recent increments of dry periods in this basin. It is necessary to consider that these studies are carried out on a local scale and in many cases the extrapolation of the results on a global scale is complicated. Future studies should be carried out in this line from a monthly point of view since there may be high variability within the seasons.

*Data avalilability*. Evapotranspiration and agrometeorological data are from the Agroclimatic Information System for Irrigation (SIAR) belonging to the Ministry of Agriculture, Fisheries and Food. These data are availability at http://eportal.mapa.gob.es/websiar/Inicio.aspx. The processing workflow for these data can see in the Material and Methods section of this paper.

*Author contributions*. RM and JA developed the idea of research and methodology and prepared the draft paper. RB and AS obtained and processed the raw data. AMT and RM prepared the maps and analyzed the statistical variables obtained. RM, JA, AS and AMT reviewed and edited the paper and contributed to the final manuscript.

*Competing interest.* The authors declare that they have no conflict of interest.

*Special issue statement.* This article is part of the special issue"Remote sensing, modelling-based hazard and risk assessment, and management of agro-forested ecosystem" It is not associated with a conference.

*Acknowledgements.* Special thanks are due to the Centro de Estudios e Investigación para la gestión de Riesgos Agrarios y Medioambientales (CEIGRAM). Also, we would like to acknowledge to the referees and especially the editor for they valuable comments and efforts in reviewing and handing our paper.

*Financial support.* This research has been supported by MINECO (Ministerio de Economía y Competitividad) through project PRECISOST (AGL2016-77282-C3-2-R) and project AGRISOST-CM (S2018/BAA-4330).

*Review statement.* This paper was edited by Jonathan Rizzi and rewiewed by Pankaj Pandey and ############

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
