# Peer review of "Estimation of evapotranspiration by FAO Penman-Monteith Temperature and Hargreaves-Samani models under temporal and spatial criteria. A case study in Duero Basin (Spain)."

_Natural Hazards and Earth System Sciences, 2019_

## Referee Comment (RC1) · Anonymous Referee #1 · 4 Sep 2019

In this paper, the authors evaluated two temperature-based methods (PMT and HS) to estimate evapotranspiration under spatial and temporal criteria, in the Duero basin (Spain). For ameliorating the document, the following suggestions are proposed for changes: 1. Add information on the quality of the data and which techniques they used to detect outliers and for the filling of data.

2. On line 212, it indicates that the temperature was used to estimate the wind speed, when they only actually used the average or set the value of 2 m / s.

3. On line 287, rewrite the paragraph in a more understandable way: RMSE is 0.55 for the PMTOUH model.

4. In the conclusion. Please provide also the limitation and future studies of this research.

5. Manuscript needed some language polishing; technical errors exist in the manuscript. Please improve them to strengthen the readership of journal.

I hope these comments will be helpful to you. My sense of the reviewers' comments is that there is a very good basis on which I can recommend that this paper be modified in a responsive manner to the comments above. If the modification is done carefully and completely, upon re-submission and evaluation, I think you can be confident that the paper will be accepted for publication.

---

## Referee Comment (RC2) · Anonymous Referee #2 · 4 Sep 2019

General comments - This paper feeds into an important topic where a changing climate and increasing population makes the current and future efficient use of water resources essential. The consideration of the performance of various evapotranspiration (ET) models requiring readily available data inputs, in comparison to the standard FAO56 Penman-Monteith reference ET model is therefore of value for the chosen semi-arid location where irrigation is used. The paper is generally well presented using clear figures and tables, however some of the phraseology needs to be improved. It is my opinion that the paper is worthy of publication following minor amendments.

[Figure]

Specific comments - The scientific significance is good. The Hargreaves-Samani (HS) model has been shown to have acceptable levels of accuracy in other arid regions of the world, however, the results showing that the simple calibrated HS (HSC) model performed well in spatial and temporal comparisons to other calculation methods will be of value in irrigation planning in this region of high agricultural water use. The scientific quality is good with valid and generally clear methods. However, as minor points: the methods section also needs to include the time-step used for the ET calculations; the paragraph from lines 205 to 211 would be better placed in the Introduction; and it is not clear whether the first few lines of the Results and discussion section relate to a general site description or are for the study period of the paper. The presentation quality is fair but this could be easily improved (see Technical corrections below for details). The use of the word 'reality' in line 406 is not appropriate, as the comparison in the paper is to reference ET not actual ET. In reality ET will be different to the ETO due to a number of factors e.g. crop type. The table and figures are good, being both appropriate and readable.

Technical corrections - Most of these are small and easily rectified. I have not provided an exhaustive list (please re-check the document carefully), but the following stand out as needing to be improved: Line 40, I would rephrasing the first line to start with 'A growing population. . .'; Lines 46-47, 'represented as a loss' would be better phrased as 'represents a loss'; Line 51, 'allows calculating' may be better as 'that allows the calculation of'; Lines 61-63 starting 'ETo is affecting' are not clear. Perhaps this should be 'ETo is only affected by climatic parameters, and is computed from weather data. Crop influences are accounted for by using a specific crop coefficient (KC). '; Line 64, would be better as '. . . because of climate differences'; Line 68, I'm not sure what 'campaign' refers to here. Maybe planting or the study period? This needs clarification in the text; Line 95 would be better as '. . . the annual calibration being the most studied.'; The paragraph starting at line 100 should mention that the ET models are evaluated against the FAO56 Penman-Monteith model; Lines 120-120 would be improved by 'However, precipitation ranges from minimum values of 400 . . ..to a maximum of 1800 . . .'; Line

133, it is not clear what the 10% refers to here; Line 161, 'incorporate' should be 'incorporates'; Line 173 should start 'The FAO . . .'; Line 174 should be 'the Penman-Monteith . . .' and '. . . temperature-based models'; Line 184, the end of the sentence and beginning of the next contain typographic errors; Line 191, would be better as 'to calibrate . . .'; Line 193, it would be good to mention here that the calculation of Ho is detailed further on in the paper; Line 198 would be better as 'such as topography, . . . (among others) thus using a fixed . . .'; Line 273, the abbreviations for the months (e.g. DJF) need to be expanded on first use in the main text; Line 284, I would rephrase 'best behaviour'; Line 285, 'shows no tendency', this is not clear i.e. no tendency to what?; Lines 288-289 are not clearly worded and need to be re-written; Lines 289-290, 'Respect to the models. . .' should be replaced by something like 'The performance of the models [specify which models] improve as the averages of . . . .' ; Lines 299-300, 'showed for the PMT model better performance than for the Hargreaves. . .' would be better as 'showed the PMT model performed better than the Hargreaves . . .'; Line 325, the abbreviation 'DPV' is not expanded on first use in the main text; Lines 369-376 are not easy to follow and would benefit from rewriting; Line 428, '. . . winter is a season that does not worry too much' should be rephrased; Line 429, it is not clear what the 1 mm refers to (ET, I presume), the whole sentence needs to be more clearly written; Line 431, 'season' should be 'seasonal'.

---

## Referee Comment (RC3) · Anonymous Referee #3 · 5 Sep 2019

The article reports a complete work on the estimation of reference evapotranspiration using Hargreaves and Temperature Penman-Monteith FAO56 equations, introducing calibration in both models. On the basis of the subject matter, the paper falls within the general scope of the Natural and Earth System Sciences Journal. Overall the paper was fairly well written, and it is interested to the Journal readers. The abstract is sufficiently informative. The introduction is well elaborated and documented by numerous and significant references. Materials and methods include a detailed description of the measurements and methods used in the work. Finally, results are sounds and justified

by the outputs presented in the paper (tables and figures). We advise to introduce some recommendations that would improve the manuscript. Considering that the main source of information is a meteorological database, a detailed explanation of the quality control procedures and validation of the meteorological data used in the study would be necessary. In addition, the model calibration section is too concise, and it would be necessary to detail the procedure properly. It would be advisable to include, in addition, an indicator of the performance of the models such as the relative error, ratio between the root mean square error and the average value of the measured variable. The authors do not adequately assess the good behaviour of the Hargreaves-Samani equation in its original version. In many cases, the improvement obtained after the calibration of the model is very small. It is advisable to quantify the improvement that occurs in each of the models after calibration. Finally, a weakness of the paper is that it presents too many results and in many cases a lack of discussion and comparison with other results of similar works. I recommend a Discussion section independent of the Results.

---

## Referee Comment (RC4) · Anonymous Referee #4 · 21 Sep 2019

This study presents a calibration of Hargreaves evapotranspiration models. The study borrows its fundamental from numerous published studies on similar work, which present almost the same method. Although the level of novelty is not high, the paper does present an interesting analysis and is an interesting issue in the chosen problem. Thus, the paper can be considered for publication provided the following issues are addressed:

Abstract: What is PMTCUH, PMTOUH ? Authour needs to define these at its first use. The abstract should be revised. In my opinion, it is not necessary to present the values

for performance evaluation of fitted models. If you have to show the difference in performance of fitted models, you should note to the performance evaluation of seasonal scale also between annual and seasonal scale.

Introduction: The introduction needs to sharpened. The justification of the study needs to explains how this work is different from many other similar published studies like " Pandey et al (2014) Calibration and performance verification of Hargreaves Samani equation in a humid region. Irrigation and Drainage 63(5): 659-667. DOI: 10.1002/ird.1874 and Pandey, P.K. & Pandey, V(2016) Evaluation of temperature-based Penman–Monteith (TPM) model under the humid environment Model. Earth Syst. Environ. (2016) 2: 152. https://doi.org/10.1007/s40808-016-0204-9 . In this regard, I suggest that you refer to above mentioned studies in order to improve justification of the study.

Materials and Methods The description of study area needs to shortened. The main approach of this study to improve Hargreaves model is based on calibrations of Krs coefficient. However, improvement also possible by calibrating exponent of the Eq. Justification need to explained in this regard. In evaluation of models performance either intercomparing of indices should discussed or author use composite index. The advantages of composite index is that all the selected indices were normalized between 0 and 1 to avoid the potent stimulus of any particular index. Due to this, maxima value of any index is scaled to 1 and minima value to 0 (Pandey & Pandey (2018); doi: 10.2166/wcc.2018.305).

Results & Discussion: The main problem with this section is poor discussion. I suggest author add separate discussion section to improve presentation of results. Also, if possible, add composite index as used by Pandey & Pandey (2018, doi: 10.2166/wcc.2018.305) in evapotranspiration study. Conclusion: As conclusion section is dependent on results and discussion section. In my view author first revise result and discussion section. Afterwards present only core finding in conclusion section.

---

## Author Comment (AC1) · 12 Nov 2019

We thank Reviewer1 for the many insightful comments and suggestions. Next, we respond carefully to all suggestions in the attached file. We also include the revised version of the manuscript (NHESS-2019-250).

Please also note the supplement to this comment: https://www.nat-hazards-earth-syst-sci-discuss.net/nhess-2019-250/nhess-2019-250-AC1-supplement.zip

---

## Author Comment (AC2) · 12 Nov 2019

We thank Reviewer2 for the many insightful comments and suggestions. Next, we respond carefully to all suggestions in the attached file. We also include the revised version of the manuscript (nhess-2019-250).

Please also note the supplement to this comment:
https://www.nat-hazards-earth-syst-sci-discuss.net/nhess-2019-250/nhess-2019-250-AC2-supplement.zip

---

## Author Comment (AC3) · 12 Nov 2019

We thank Reviewer3 for the many insightful comments and suggestions. Next, we respond carefully to all suggestions in the attached file. We also include the revised version of the manuscript (nhess-2019-250).

Please also note the supplement to this comment:
https://www.nat-hazards-earth-syst-sci-discuss.net/nhess-2019-250/nhess-2019-250-AC3-supplement.zip

---

## Author Response (AR1)

Authors' responses to review comments are in **red, bold**

**Anonymous Referee #1**

In this paper, the authors evaluated two temperature-based methods (PMT and HS) to estimate evapotranspiration under spatial and temporal criteria, in the Duero basin (Spain). For ameliorating the document, the following suggestions are proposed for changes: 1. Add information on the quality of the data and which techniques they used to detect outliers and for the filling of data. **OK., Done**

2. On line 212, it indicates that the temperature was used to estimate the wind speed, when they only actually used the average or set the value of 2 m / s. **We agree. It has been changed**

3. On line 287, rewrite the paragraph in a more understandable way: RMSE is 0.55 for the PMTOUH model. **Modified**

4. In the conclusion. Please provide also the limitation and future studies of this research. **OK, Done**

5. Manuscript needed some language polishing; technical errors exist in the manuscript. Please improve them to strengthen the readership of journal. I hope these comments will be helpful to you. **Done.** My sense of the reviewers' comments is that there is a very good basis on which I can recommend that this paper be modified in a responsive manner to the comments above. If the modification is done carefully and completely, upon re-submission and evaluation, I think you can be confident that the paper will be accepted for publication.

Authors' responses to review comments are in **red, bold**

**Anonymous Referee #2**

*General comments* - This paper feeds into an important topic where a changing climate and increasing population makes the current and future efficient use of water resources essential. The consideration of the performance of various evapotranspiration (ET) models requiring readily available data inputs, in comparison to the standard FAO56 Penman-Monteith reference ET model is therefore of value for the chosen semi-arid location where irrigation is used. The paper is generally well presented using clear figures and tables, however some of the phraseology needs to be improved. It is my opinion that the paper is worthy of publication following minor amendments.

**Thank you so much for your summary in our work and the importance ot it.**

*Specific comments* - The scientific significance is good. The Hargreaves-Samani (HS) model has been shown to have acceptable levels of accuracy in other arid regions of the world, however, the results showing that the simple calibrated HS (HSC) model performed well in spatial and temporal comparisons to other calculation methods will be of value in irrigation planning in this region of high agricultural water use. The scientific quality is good with valid and generally clear methods. However, as minor points: the methods section also needs to include the time-step used for the ET calculations; the paragraph from lines 205 to 211 would be better placed in the Introduction **(OK, Done);** and it is not clear whether the first few lines of the Results and discussion section relate to a general site description or are for the study period of the paper. The presentation quality is fair but this could be easily improved (see Technical corrections below for details). The use of the word 'reality' in line 406 is not appropriate, as the comparison in the paper is to reference ET not actual ET **(OK).** In reality ET will be different to the ETO due to a number of factors e.g. crop type. The table and figures are good, being both appropriate and readable.

***Technical corrections*** - Most of these are small and easily rectified. I have not provided an exhaustive list (please re-check the document carefully), but the following stand out as needing to be improved:

Line 40, I would rephrasing the first line to start with 'A growing population. . .'; **Done**

Lines 46-47, 'represented as a loss' would be better phrased as 'represents a loss'; **Ok, Corrected.**

Line 51, 'allows calculating' may be better as 'that allows the calculation of'; **Ok, Done**

Lines 61-63 starting 'ETo is affecting' are not clear. Perhaps this should be 'ETo is only affected by climatic parameters, and is computed from weather data. Crop influences are accounted for by using a specific crop coefficient (KC). ';**Ok paragraph modified**.

Line 64, would be better as '. . . because of climate differences'; **Done**

Line 68, I'm not sure what 'campaign' refers to here. Maybe planting or the study period? This needs clarification in the text; **OK, campaign has been replaced by cultivation period**

Line 95 would be better as '. . . the annual calibration being the most studied.'; **We agree, Done**

The paragraph starting at line 100 should mention that the ET models are evaluated against the FAO56 Penman Monteith model; **We agree, Corrected**

Lines 120-120 would be improved by 'However, precipitation ranges from minimum values of 400 . . ..to a maximum of 1800 . . .'; **Modified**

Line 133, it is not clear what the 10% refers to here; **Corrected**

Line 161, 'incorporate' should be'incorporates'; **Corrected**

Line 173 should start 'The FAO . . .'; Line 174 should be 'the Penman-Monteith . . .' and '. . . temperature-based models'; **OK**

Line 184, the end of the sentence and beginning of the next contain typographic errors; **OK Corrected**

Line 191, would be better as 'to calibrate . . .';**Ok Done**

Line 193, it would be good to mention here that the calculation of Ho is detailed further on in the paper; **It Could be considered but we consider that in this paragraph we are describing equation (2). So we don't see it necessary**.

Line 198 would be better as 'such as topography, . . . (among others) thus using a fixed . . .'; **Ok Done**

Line 273, the abbreviations for the months (e.g. DJF) need to be expanded on first use in the main text; **I agree with the reviewer, they were described above on lines 252 and 253**

Line 284, I would rephrase 'best behaviour'; **OK.**

Line 285, 'shows no tendency', this is not clear i.e. no tendency to what?; **OK the phrase has been rewording to explain it better**

Lines 288-289 are not clearly worded and need to be re-written; **Ok modified**

Lines 289 290, 'Respect to the models. . .' should be replaced by something like 'The performance of the models [specify which models] improve as the averages of . . ..' ; **OK, Done**

Lines 299-300, 'showed for the PMT model better performance than for the Hargreaves. . .' would be better as 'showed the PMT model performed better than the Hargreaves . . .'; **OK Corrected**

Line 325, the abbreviation 'DPV' is not expanded on first use in the main text; **OK, Changed**

Lines 369-376 are not easy to follow and would benefit from rewriting;

Line 428, '. . . winter is a season that does not worry too much' should be rephrased; **Ok, Done**

Line 429, it is not clear what the 1 mm refers to (ET, I presume), the whole sentence needs to be more clearly written; **Modified**

Line 431, 'season' should be 'seasonal'. **Done**

Authors' responses to review comments are in **red, bold**

**Anonymous Referee #3**

The article reports a complete work on the estimation of reference evapotranspiration using Hargreaves and Temperature Penman-Monteith FAO56 equations, introducing calibration in both models. On the basis of the subject matter, the paper falls within the general scope of the Natural and Earth System Sciences Journal. Overall the paper was fairly well written, and it is interested to the Journal readers. The abstract is sufficiently informative. The introduction is well elaborated and documented by numerous and significant references. Materials and methods include a detailed description of the measurements and methods used in the work. Finally, results are sounds and justified by the outputs presented in the paper (tables and figures). We advise to introduce some recommendations that would improve the manuscript.

**Thank you for your time and work on this manuscript.**

Considering that the main source of information is a meteorological database, a detailed explanation of the quality control procedures and validation of the meteorological data used in the study would be necessary (**Done**). In addition, the model calibration section is too concise, and it would be necessary to detail the procedure properly. It would be advisable to include, in addition, an indicator of the performance of the models such as the relative error, ratio between the root mean square error and the average value of the measured variable. (**We included the mean value of daily ETo and the RMSE (mm·d-1) in table 3, we consider that the relative error (%) can be obtained indirectly,  but we agree with the review and for a better understanding  we have included RMSE (%) in table 3**). The authors do not adequately assess the good behaviour of the Hargreaves-Samani equation in its original version. In many cases, the improvement obtained after the calibration of the model is very small. It is advisable to quantify the improvement that occurs in each of the models after calibration. **OK Included in the text in the paragraph below Table 3.** Finally, a weakness of the paper is that it presents too many results and in many cases a lack of discussion and comparison with other results of similar works. I

recommend a Discussion section independent of the Results. **OK, we have separated both.**

Authors' responses to review comments are in **red, bold**

**Anonymous Referee #4**

This study presents a calibration of Hargreaves evapotranspiration models. The study borrows its fundamental from numerous published studies on similar work, which present almost the same method. Although the level of novelty is not high, the paper does present an interesting analysis and is an interesting issue in the chosen problem.

**Thank you for your interest in this work.**

Thus, the paper can be considered for publication provided the following issues are addressed:

**Abstract**: What is PMTCUH, PMTOUH ? Author needs to define these at its first use., **OK Done.**

The abstract should be revised. In my opinion, it is not necessary to present the values for performance evaluation of fitted models. If you have to show the difference in performance of fitted models, you should note to the performance evaluation of seasonal scale also between annual and seasonal scale.

**We have review and reword the Abstract.**

**Introduction**: The introduction needs to sharpened. The justification of the study needs to explains how this work is different from many other similar published studies like "Pandey et al (2014) Calibration and performance verification of Hargreaves Samani equation in a humid region. Irrigation and Drainage 63(5): 659-667. DOI: 10.1002/ird.1874 and Pandey, P.K. & Pandey, V(2016) Evaluation of temperature based Penman–Monteith (TPM) model under the humid environment Model. Earth Syst. Environ. (2016) 2: 152. https://doi.org/10.1007/s40808-016-0204-9 . In this regard, I suggest that you refer to above mentioned studies in order to improve justification of the study. **We have reviewed the articles and have included in the introduction of the articles by Pandey and Pandey (2016) and Pandey et al. (2014).**

**Materials and Methods**: The description of study area needs to shortened. **Our study focuses on the Duero Basin. We believe that the calibrations and models tested depend on the study area. The extrapolation of the data in many cases is very difficult and depends on the characteristics of the area.**

**The detailed description of the area will guide if possible the extrapolation of the results so we consider this part important. On the other hand, the estimation of ETo in the area is very important due to the high agricultural activity, in our view a vision of this activity is important for the optimization of water use**. The main approach of this study to improve Hargreaves model is based on calibrations of Krs coefficient. However, improvement also possible by calibrating exponent of the Eq. **The authors Hargreaves and Samani point out that the krs coefficient needs to be adjusted, and subsequently developed procedures to adjust the krs coefficient (Z. Samani, J. of Irrig. & Drainage Engr., 126(4). Analyzing the applications of the HS equation, Hargreaves and Allen (2003) concluded that "recalibrating the exponents and coefficients of the HS equation only increased the complexity of the equation". Very good results were reported by Todorovic et al. (2013) and Raziei and Pereira (2013a) relative to calibrating kRs for a wide range of climates.**

Justification need to explained in this regard. In evaluation of models performance either intercomparing of indices should discussed or author use composite index. The advantages of composite index is that all the selected indices were normalized between 0 and 1 to avoid the potent stimulus of any particular index. Due to this, maxima value of any index is scaled to 1 and minima value to 0 (Pandey & Pandey (2018); doi: 10.2166/wcc.2018.305). **Pandey et al. (2018) used a Weighted root mean square error (WRMSE). The index is calculated based on the combined influence of both RMSE and adjusted RMSE (ARMSE). Also used the Global performance index (GPI) for final ranking. The GPI is based on the assumption that if the value of the indicator is higher than the median, then the higher the difference between the two reduces the accuracy of the equation. This index are very interesting and they will be considered in future research.**

*Results & Discussion:* The main problem with this section is poor discussion. I suggest author add separate discussion section to improve presentation of results **Ok, Done** . Also, if possible, add composite index as used by Pandey & Pandey (2018, doi: 10.2166/wcc.2018.305) in evapotranspiration study. **This paper uses the performance indexes most used in the topic; and allow us to properly validate the results. It will be considered in future research.**

Conclusion: As conclusion section is dependent on results and discussion section. In my view author first revise result and discussion section. Afterwards present only core finding in conclusion section.

[revised manuscript text omitted]

---

## Author Response (AR2)

Dear Editor,

As you can see in the manuscript and according to the Referee#1 (Report#2)  we have made the two small changes indicated to us:

*"1. On line 328 where it says, "the best behavior is shown by the PMT CHU..." the sub index should be CUH and not CHU.*

*2. In table 3, in the last column, it should be ET0 FAO56, mm.d-1."*

We have detected that in figure 7 the letter G of a map was omitted and we have include this letter.

We have change Fig. by Figure in Figure 5, 6 and 7.

Bests Regards